

# Assessment of antimycobacterial activities of pure compounds extracted from Thai medicinal plants against clarithromycin-resistant *Mycobacterium abscessus*

Auttawit Sirichoat[1,2,*], Irin Kham-ngam[1,2,*], Orawee Kaewprasert[1,2], Pimjai Ananta[1,3], Awat Wisetsai[4], Ratsami Lekphrom[4] and Kiatichai Faksri[1,2]

[1] Department of Microbiology, Faculty of Medicine, Khon Kaen University, Khon Kaen, Thailand
[2] Research and Diagnostic Center for Emerging Infectious Diseases, Khon Kaen University, Khon Kaen, Thailand
[3] Clinical Laboratory Unit, Srinagarind Hospital, Faculty of Medicine, Khon Kaen University, Khon Kaen, Thailand
[4] Natural Products Research Unit, Department of Chemistry, and Center for Innovation in Chemistry, Faculty of Science, Khon Kaen University, Khon Kaen, Thailand
* These authors contributed equally to this work.

Corresponding author
Kiatichai Faksri, kiatichai@kku.ac.th

## ABSTRACT

**Background:** Infection with *Mycobacterium abscessus* is usually chronic and is associated with clarithromycin resistance. Increasing drug resistance is a major public-health problem and has led to the search for new antimycobacterial agents. We evaluated the antimycobacterial activity, toxicity, and synergistic effects of several plant secondary metabolites against *M. abscessus*.

**Methods:** Twenty-three compounds were evaluated for antimycobacterial activity against thirty *M. abscessus* clinical isolates by broth microdilution to determine their minimum inhibitory concentration (MIC) values. Toxicity was evaluated using red and white blood cells (RBCs and WBCs). The compounds were used in combination with clarithromycin to investigate the possibility of synergistic activity.

**Results:** Five out of twenty-three compounds (RL008, RL009, RL011, RL012 and RL013) exhibited interesting antimycobacterial activity against *M. abscessus*, with MIC values ranging from <1 to >128 μg/mL. These extracts did not induce hemolytic effect on RBCs and displayed low toxicity against WBCs. The five least-toxic compounds were tested for synergism with clarithromycin against seven isolates with inducible clarithromycin resistance and seven with acquired clarithromycin resistance. The best synergistic results against these isolates were observed for RL008 and RL009 (8/14 isolates; 57%).

**Conclusions:** This study demonstrated antimycobacterial and synergistic activities of pure compounds extracted from medicinal plants against clarithromycin-resistant *M. abscessus*. This synergistic action, together with clarithromycin, may be effective for treating infections and should be further studied for the development of novel antimicrobial agents.

# INTRODUCTION

Nontuberculous mycobacteria (NTM) are found in the environment (soil and water) (*Velayati et al., 2014*). Some species of NTM can cause life-threatening human diseases with a high mortality rate (*Cassidy et al., 2009*; *Iroh Tam et al., 2015*). *Mycobacterium abscessus* is a common NTM species causing chronic infection and is highly associated with drug resistance (*Tung et al., 2015*). Drug-resistant *M. abscessus* infection has become a serious health issue in many countries, including Thailand (*Kham-Ngam et al., 2018*). Treatment of *M. abscessus* infection is prolonged and one-third of cases are associated with treatment failure (*Nessar et al., 2012*). Although clarithromycin is a drug of choice, half of the strains present in Thailand are clarithromycin-resistant (*Ananta et al., 2018*). Therefore, new treatment alternatives are needed to overcome drug-resistant *M. abscessus* infection.

Plants are a source of bioactive compounds that can treat various diseases (*Rios & Recio, 2005*). Several research teams have reported anti-*Mycobacterium tuberculosis* activity of extracts from *Tetradenia riparia* (*Baldin et al., 2018*), *Persea americana* (*Jimenez-Arellanes et al., 2013*), *Lophira lanceolata* (*Nkot et al., 2018*) and *Flourensia cernua* (*Molina-Salinas et al., 2006*). Northeastern Thailand has high plant diversity, which remains locally important as a source of traditional medicines (*Kaewpiboon et al., 2012*). In Thailand, extracts from *Neonothopanus nambi* (*Kanokmedhakul et al., 2012*) and *Rothmannia wittii* (*Chaipukdee et al., 2016*) showed antimycobacterial activity against *M. tuberculosis*. Atalantiaphyllines A–G, isolated from roots of *Atalantia monophylla*, exhibited higher aromatase inhibition than did ketoconazole, and also showed high α, α-diphenyl-β-picrylhydrazyl (DPPH) radical-scavenging activity (*Pailee et al., 2020*). Anthracene and anthraquinone metabolites isolated from *Prismatomeris filamentosa* showed antibacterial activities against Gram-positive and Gram-negative bacteria such as *Bacillus subtilis*, *Bacillus cereus*, *Staphylococcus aureus*, *Escherichia coli*, *Pseudomonas aeruginosa* and *Shigella sonnei* (*Wisetsai, Lekphrom & Schevenels, 2021*). The compounds derived from luminescent mushroom *Neonothopanus nambi* exhibited antimalarial activity against *Plasmodium falciparum* (*Kanokmedhakul et al., 2012*). However, the effect of these plant extracts against *M. abscessus* has not been reported.

We aimed to evaluate the antimycobacterial activities of pure compounds extracted from four medicinal plants (*A. monophylla*, *P. filamentosa*, *Ageratum conyzoides* and *R. wittii*) and from the cultured mycelium of the luminescent mushroom *N. nambi*. These compounds were tested against clarithromycin-susceptible and non-susceptible *M. abscessus* clinical isolates. The toxicity for mammalian cells and synergistic effects of selected compounds with clarithromycin were also analyzed.
**Table 1 Compounds used in this study.**

| No. | Code | Compound | Source | References |
|---|---|---|---|---|
| 1 | RL001 | *N*-methylcycloatalaphylline A | Roots of *A. monophylla* | (*Pailee et al., 2020*) |
| 2 | RL006 | yukocitrine | Roots of *A. monophylla* | (*Pailee et al., 2020*) |
| 3 | RL002 | *N*-methylatalaphylline | Roots of *A. monophylla* | (*Pailee et al., 2020*) |
| 4 | RL007 | atalaphylline | Roots of *A. monophylla* | (*Pailee et al., 2020*) |
| 5 | RL004 | atalaphylline-3,5-dimethyl ether | Roots of *A. monophylla* | (*Pailee et al., 2020*) |
| 6 | RL003 | 2,2-dimethylchromenocoumarin | Roots of *A. monophylla* | (*Pailee et al., 2020*) |
| 7 | RL005 | auraptene | Roots of *A. monophylla* | (*Pailee et al., 2020*) |
| 8 | RL009 | rubiadin-1-methyl ether | Roots of *P. filamentosa* | (*Wisetsai, Lekphrom & Schevenels, 2021*) |
| 9 | RL010 | rubiadin | Roots of *P. filamentosa* | (*Wisetsai, Lekphrom & Schevenels, 2021*) |
| 10 | RL011 | knoxiadin | Roots of *P. filamentosa* | (*Wang, Chen & Lu, 1985*) |
| 11 | RL008 | nordamnacanthal | Roots of *P. filamentosa* | (*Wisetsai, Lekphrom & Schevenels, 2021*) |
| 12 | RL012 | damnacanthal | Roots of *P. filamentosa* | (*Wisetsai, Lekphrom & Schevenels, 2021*) |
| 13 | RL013 | damnacanthol | Roots of *P. filamentosa* | (*Wisetsai, Lekphrom & Schevenels, 2021*) |
| 14 | RL014 | 3′,4′,7-tri-*O*-methylluteolin | Flowers of *A. conyzoides* | (*Ahond et al., 1990*) |
| 15 | RL015 | 4′,7-di-*O*-methylapigenin | Flowers of *A. conyzoides* | (*Ahond et al., 1990*) |
| 16 | RL016 | 4′-*O*-methylapigenin | Flowers of *A. conyzoides* | (*Yim, Kim & Lee, 2003*) |
| 17 | RL017 | 2′-hydroxy-4,4′,6′-trimethoxychalcone | Flowers of *A. conyzoides* | (*Sukari et al., 2004*) |
| 18 | RL020 | 3,5-dihydroxycinnamate | Roots of *R. wittii* | (*Wisetsai et al., 2020*) |
| 19 | RL021 | lippianoside B | Roots of *R. wittii* | (*Wisetsai et al., 2020*) |
| 20 | RL022 | rothmannioside C | Roots of *R. wittii* | (*Wisetsai et al., 2020*) |
| 21 | RL023 | rothmannioside A | Roots of *R. wittii* | (*Wisetsai et al., 2020*) |
| 22 | RL024 | rothmannioside B | Roots of *R. wittii* | (*Wisetsai et al., 2020*) |
| 23 | RL019 | aurisin A | Cultured mycelium of *N. nambi* | (*Kanokmedhakul et al., 2012*) |

**Note:**
Compounds **1–22** were isolated from four medicinal plants (*Atalantia monophylla*, *Prismatomeris filamentosa*, *Ageratum conyzoides*, and *Rothmannia wittii*). Compound **23** was isolated from the cultured mycelium of the luminescent mushroom *Neonothopanus nambi*.

## MATERIALS AND METHODS

### Pure compounds extracted from local medicinal plants

A total of 23 pure compounds, 22 isolated from four medicinal plants (*A. monophylla*, *P. filamentosa*, *A. conyzoides* and *R. wittii*) and one compound isolated from the cultured mycelium of a luminescent mushroom (*N. nambi*), were evaluated (Table 1 and Fig. S1). The compounds were purified using column chromatography techniques to isolate the pure secondary metabolites, as described previously (*Sombatsri et al., 2018*). Stock solutions were prepared by dissolving pure compounds in dimethyl sulfoxide (DMSO), which can dissolve both polar and non-polar compounds.

### Bacterial isolates

*Mycobacterium abscessus* clinical isolates were retrieved from the culture collection of Srinagarind Hospital, Khon Kaen University, Thailand. The colony morphology of each isolate was noted. The identification of *M. abscessus* species was performed according to the protocol previously published (*Kham-Ngam et al., 2019*). All isolates were

sub-cultured on Lowenstein-Jensen (LJ) media and incubated at 37 °C for 7 days before further analysis. Informed consent was not required for this study. Anonymized, left-over specimens were used. All specimens, including isolates and blood samples, had been obtained during routine practice. This study was approved by the Khon Kaen University Ethics Committee for Human Research (HE611496).

## Antibiotic susceptibility testing

The minimum inhibitory concentration (MIC) for clarithromycin was determined according to published protocols (*Kham-Ngam et al., 2019*). The broth-microdilution method using a RAPMYCOI Sensititre 96-well plate (TREK Diagnostic Systems, Independence, OH, USA) following the manufacturer's protocol. Briefly, individual colonies of *M. abscessus* were suspended in demineralized water to obtain a density corresponding to McFarland Standard 0.5. Then, 50 μL of cell suspension were transferred into a tube of cation-adjusted Mueller-Hinton broth (TREK Diagnostic Systems, Independence, OH, USA) with TES buffer (to optimize the conditions for antibiotic stability) to achieve a final cell concentration of approximately $5 \times 10^5$ CFU/mL. One-hundred microliters of this inoculum were added to each well of a 96-well plate containing different concentrations of antibiotics. The 96-well microtiter plates were then incubated at 37 °C for 3–14 days under aerobic condition. Clarithromycin susceptibility was read at 3, 5, and 14 days according to CLSI guidelines (*CLSI, 2018*). A reading at day 3 was used to test for inducible resistance according to previously described protocols (*Ananta et al., 2018*). Inducible resistance was inferred by changes in MIC values from "susceptible" at day 3 to "resistant" at day 14. Isolates that were resistant on day 3 and thereafter were regarded as demonstrating acquired resistance. All clarithromycin-susceptible and clarithromycin-resistant (both inducible and acquired resistance) *M. abscessus* clinical isolates were used for further analysis.

## Antimycobacterial assay

The antimycobacterial assay was carried out using a broth-microdilution method to determine the MIC values according to the CLSI guidelines (*CLSI, 2018*). Two-fold serial dilutions of pure compounds were prepared directly in a 96-well microtiter plate. For preparation of the *M. abscessus* inoculum, the same protocol as described above was used. One-hundred microliters of this inoculum were mixed with 100 μL of pure compound (to give the final concentrations: 1, 2, 4, 8, 16, 32, 64 and 128 μg/mL) and were then added to each well of the 96-well plate. Following incubation for 3–5 days at 37 °C, MICs were visually determined as the lowest concentration of the compound that completely inhibited the mycobacterial growth (*Ananta et al., 2018*).

## Hemolytic assay

Hemolytic activity was determined according to protocols published previously with some modification (*Lima Viana et al., 2018*). Briefly, 6 mL of blood from a single healthy volunteer were collected and transferred into a heparin collection tube. Whole blood was

centrifuged at 5,000 rpm for 5 min and the plasma was then discarded. Concentrated red blood cells (RBCs) were isolated and washed three times with 1% sterile phosphate buffer saline (PBS) solution (pH 7.4) and centrifugation. Then, the RBCs were diluted in 1% PBS to a 5% final concentration of RBC suspension for analysis (950 μL of 1% PBS and 50 μL of concentrated RBCs in a total volume of 1 mL). Two hundred microliters of this RBC suspension were transferred into the tubes containing different concentrations of pure compounds in a total volume of 1 mL (at concentrations before adding the RBC suspension: 1, 2, 4, 8, 16, 32, 64 and 128 μg/mL in 1% PBS). Positive and negative controls were used, these being 1% Triton X-100 solution (*Lima Viana et al., 2018*) and 1% PBS, respectively. The final volume of each experiment was 1.0 mL. The solutions were incubated at 37 °C for 1 h. After incubation, the suspensions were then centrifuged at 3,000 × *g* for 2 min. Then, 100 μL of supernatant from each tube were transferred into a 96-well plate for measurement of the absorbance at 540 nm using a microplate reader (each absorbance was measured twice). In addition, RBC morphology was observed under a light microscope and recorded. All tests were performed in duplicate for each test compound. Hemolytic activity was calculated by the following equation:

$$\text{Hemolysis } (\%) = [(As - An)/(Ac - An)] \times 100$$

where *As* refers to the absorbance of the sample, *An* refers to the absorbance of the negative control (RBCs with PBS) and *Ac* refers to the absorbance of the positive control (RBCs with Triton X-100).

## Cell viability assay

To assay the toxicity of each tested compound for human white blood cells, the trypan-blue exclusion test was used (*Strober, 2015*). Briefly, white blood cells (WBCs) were isolated from 6 mL of blood from a healthy volunteer using the Ficoll density-gradient technique (*Boyum, 1976*). Whole blood was carefully transferred into a tube containing Ficoll solution (ratio 1:1). Then, the cells were centrifuged at 1,500 rpm for 10 min at 20 °C and the WBC layer was transferred into a new tube. The concentrated WBCs were washed three times with 1% PBS at 1,500 rpm for 5 min at 20 °C and re-suspended in 1 mL of RPMI-1640 media (Gibco™, New York, USA). Fifty microliters of WBC suspension were transferred into individual wells of a 96-well plate and then 50 μL of pure compounds at different concentrations (ranging from 1 to 128 μg/mL) were added. The 96-well plate was incubated at 37 °C for 1 h. Then, 20 μL of the suspension was mixed with 20 μL of 0.4% trypan blue solution in buffered isotonic salt solution (0.81% NaCl and 0.06% $K_2HPO_4$) and incubated for 3 min at room temperature. Viable and dead cells were counted under a light microscope using a hemocytometer. As a negative control, WBC suspension was treated with PBS. The test was performed in duplicate for each test compounds. Dead cells were calculated using the following equation:

$$\text{Dead cells } (\%) = (\text{number of dead cells}/\text{total number of cells}) \times 100$$

## Genome sequencing and analysis

Genomic DNA of the 30 *M. abscessus* clinical isolates was extracted using the cetyltrimethylammonium bromide-sodium chloride (CTAB) method (*De Almeida et al., 2013*) and was sent for genome sequencing (NovogeneAIT, Hong Kong) using an Illumina HiSeq platform generating 150-bp paired-end reads.

The quality of raw sequences was checked using FastQC version 0.11.7 (*Andrews, 2010*). Trimmomatic (v0.36) software (*Bolger, Lohse & Usadel, 2014*) was used to remove low-quality reads. High-quality paired-end reads were then mapped to the *M. abscessus* ATCC 19977 reference genome (GenBank accession number CU458896.1) using BWA-mem (v.0.7.17) (*Li, 2013*). For converting SAM to BAM format, sorting and indexing the bam files, SAMtools v0.1.19 algorithm was used (*Li et al., 2009*). GATK version 4.0.5. (*McKenna et al., 2010*) was used for realignment, generating coverage statistics and mapping details. Both GATK and SAMtools were used for variant calling and filtering, including single-nucleotide polymorphisms (SNPs) and small indels (*Li et al., 2009*; *McKenna et al., 2010*).

For phylogenetic analysis, a WGS-based phylogeny was analyzed using mpileup, VCF and coverage files. Maximum-likelihood analysis was performed using MEGA-7 (*Qasim et al., 2018*) with the general time-reversible (GTR) and gamma model. Support for individual nodes was assessed using 1,000 bootstrap replicates. The phylogenetic tree was visualized using iTol software (https://itol.embl.de/).

## Synergism

Combinations of plant secondary metabolites and clarithromycin were evaluated using a microdilution checkerboard method (*Garcia, 2010*). Five concentrations of each test compound (ranging from 32 to 512 μg/mL) and eight concentrations of clarithromycin (Sigma-Aldrich, St. Louis, MO, USA) (ranging from 32 to 4,096 μg/mL) were prepared, and *M. abscessus* cell suspensions, prepared as for previous experiments, were used. Fifty microliters of each pure compound and clarithromycin were mixed in a 96-well plate, and 100 μL of inoculum were then added (final concentration of each pure compound ranged from 8 to 128 μg/mL and of clarithromycin ranged from 4 to 1,024 μg/mL). The plate was incubated at 37 °C for 7 days (for isolates with acquired resistance) or 14 days (for isolates with inducible resistance). The results were recorded and interpreted as the fractional inhibitory concentration index (FICI) (*Doern, 2014*). The FICI value was calculated using the following equation:

$$FICI = [A]/(A) + [B]/(B)$$

where [A] refers to MIC (A) in combination with (B), (A) refers to MIC (A) alone, [B] refers to MIC (B) in combination with (A), and (B) refers to MIC (B) alone.

FICI values of ≤0.5, >0.5–1.0, >1.0–4.0 and >4.0 were interpreted as indicating "synergy", "additive", "indifference" and "antagonism", respectively.

## Data analysis

All quantitative data are reported as means. Comparison of quantitative data among groups of the toxicity assays was performed using one-way ANOVA followed by post-hoc

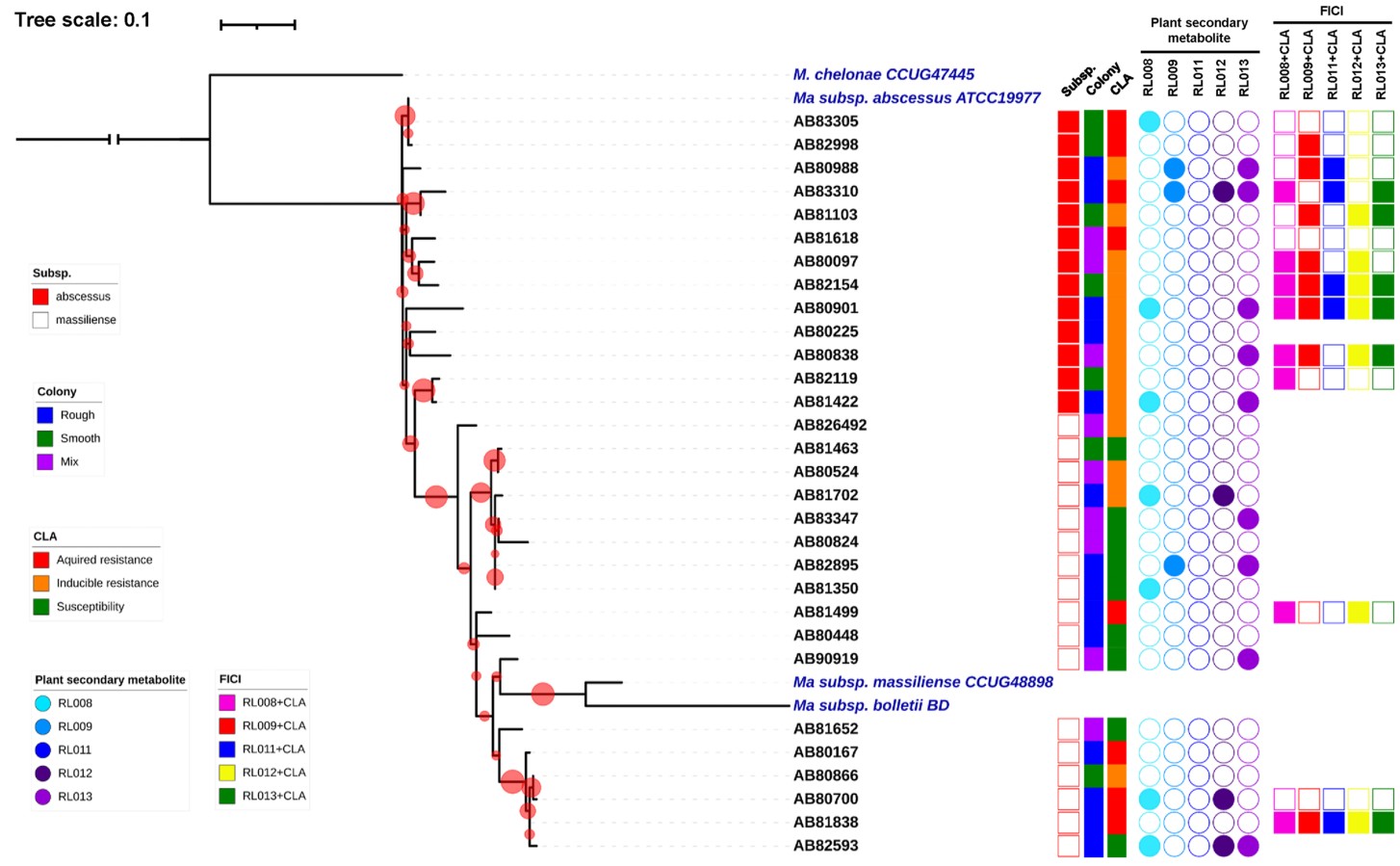

**Figure 1 Characteristics of 30 *Mycobacterium abscessus* isolates and antimycobacterial activities of five plant secondary metabolites.** All 30 isolates were either *M. abscessus* subspecies *abscessus* (red box) or subspecies *massiliense* (red border) based on genome analysis. A bootstrap consensus tree was inferred from 1,000 replicates. The phylogenetic tree was visualized using iTol software (https://itol.embl.de/). Red circles refer to bootstrap values and the size of each circle is proportional to its value (the largest red circle indicates a value of 100%). *Mycobacterium chelonae* was used as the outgroup and three reference strains of *M. abscessus* were included for comparison. Colony morphology was classified as rough (blue box), smooth (light green box) or mixed (purple box). Clarithromycin (CLA) susceptibility profiles showed acquired resistance (red box), inducible resistance (orange box), or susceptibility to CLA (green box). MIC values for the five compounds tested against the *M. abscessus* isolates ranged from 8–128 μg/mL (open circles) or were >128 μg/mL (shaded circles). Light blue, blue, dark blue, dark purple and purple circles refer to compounds labeled as RL008, RL009, RL011, RL012, and RL013, respectively. Synergistic activity of compounds in combination with clarithromycin is represented as no synergism (open box) and synergism (shaded box). Pink, red, dark blue, yellow and light green boxes refer to combinations RL008/CLA, RL009/CLA, RL011/CLA, RL012/CLA, and RL013/CLA, respectively.

LSD tests. Number of isolates with colony morphotypes and the susceptibility to the compounds and clarithromycin were compared using Fisher's exact test. *P*-values <0.05 were considered statistically significant. All statistical analyses were performed using SPSS version 19.0 (IBM, Armonk, NY, USA).

## RESULTS

### Identification and characteristics of *M. abscessus* isolates used

Thirty clinical isolates of *M. abscessus* on LJ solid medium were recovered. Characteristics of these isolates are described in Fig. 1. Thirteen isolates were identified as belonging to *M. abscessus* subsp. *abscessus* and 17 isolates to *M. abscessus* subsp. *massiliense* (Fig. 1).

Based on their phylogenetic relationships, these isolates were not clonal strains (Fig. 1). The MIC values for clarithromycin among the *M. abscessus* isolates ranged from 0.12 to ≥16 μg/mL. Ten isolates were phenotypically susceptible to clarithromycin (ranging from 0.12 to 2 μg/mL), while eleven and nine isolates exhibited inducible and acquired resistance to clarithromycin (both ≥16 μg/mL), respectively (Fig. 1 and Table 2).

## Antimycobacterial activity of plant secondary metabolites

The MIC values of 23 pure compounds were determined by broth microdilution against 30 clinical isolates of *M. abscessus*, including clarithromycin-susceptible and-resistant isolates. The MIC cut-off value of 128 μg/mL for at least one isolate was chosen for the selection of potentially useful test compounds. Of the 23 pure compounds, only five (RL008, RL009, RL011, RL012 and RL013) had MIC values lower than 128 μg/mL in any tested isolates. These five compounds were particularly effective in suppressing *M. abscessus*. The MIC values for these compounds ranged from <1 to >128 μg/mL (Table 2). The five effective pure compounds were selected for further analysis. No significant association was observed between colony morphotypes of *M. abcessus* isolates and their susceptibility to the test compounds and clarithromycin (*P*-value >0.05, Table S1).

## Toxicity testing on human RBCs and WBCs

The hemolytic activity of the five selected compounds (RL008, RL009, RL011, RL012 and RL013) on human RBCs was evaluated. None showed any hemolytic effect on RBCs (<1%) at the various concentrations used (at the compound concentration before adding RBC suspension: 1, 2, 4, 8, 16, 32, 64, and 128 μg/mL) (Fig. 2 and Table S2). Indeed, the percentage of hemolysis was less than in the negative controls (RBC suspension in 1% PBS that was used as the baseline). RBC morphologies under the light microscope were displayed as the cell shrinks (Fig. 3). These findings indicate that the compounds were not harmful to human RBCs.

For WBCs, the percentage of dead cells following exposure to each tested compound is presented in Fig. 4 and Table S3, and the statistical analysis together with ANOVA with LSD post-hoc multiple comparisons revealed significantly differed among compounds (Fig. 4F). The toxic effects were concentration-dependent. All five compounds caused death of 15–20% of cells at the final concentrations of 64–128 μg/mL, except for compound RL013, for which the mortality rate was significantly lower (3–8%). These results suggest that the compounds have a low toxicity towards human WBCs.

## Synergistic antimycobacterial activity of pure compounds and clarithromycin

Based on phenotypic results, fourteen *M. abscessus* isolates (seven isolates with inducible and seven isolates with acquired resistance to clarithromycin) (Table 2) were randomly selected to study the synergistic effect against these of five pure compounds in combination with clarithromycin. Results (Table 3) showed that the highest degree of synergism was observed for the RL008/CLA and RL009/CLA combinations (FICI ranging from 0.13 to

**Table 2** Antimycobacterial activity screening of five plant secondary metabolites against 30 *M. abscessus* isolates.

| Isolates | Organism | Colony morphology | MIC value of clarithromycin (µg/mL) | | | | | | | Antimycobacterial screening against *M. abscessus* isolates | | | | |
| | | | Day 3 | | Day 5 | | Day 14 | | DST interpretation | MIC value of pure compounds (µg/mL) | | | | |
| | | | MIC | Phenotype | MIC | Phenotype | MIC | Phenotype | | RL008 | RL009 | RL011 | RL012 | RL013 |
|---|---|---|---|---|---|---|---|---|---|---|---|---|---|---|
| 80097 | *M. abscessus* subsp. *abscessus* | Mixed | 0.25 | Susceptible | 8 | Resistant | 16 | Resistant | Inducible | 128 | 16 | 64 | 64 | 128 |
| 80167 | *M. abscessus* subsp. *massiliense* | Rough | ≥16 | Resistant | ≥16 | Resistant | ≥16 | Resistant | Acquired | 128 | 8 | 64 | 64 | 128 |
| 80225 | *M. abscessus* subsp. *abscessus* | Rough | 0.25 | Susceptible | 4 | Intermediate | ≥16 | Resistant | Inducible | <1 | <1 | <1 | <1 | <1 |
| 80448 | *M. abscessus* subsp. *massiliense* | Rough | 0.12 | Susceptible | 0.25 | Susceptible | 2 | Susceptible | Susceptible | 128 | 128 | 128 | 128 | 128 |
| 80524 | *M. abscessus* subsp. *massiliense* | Mixed | 0.5 | Susceptible | 2 | Susceptible | 2 | Susceptible | Susceptible | 128 | 128 | 128 | 128 | 128 |
| 80700 | *M. abscessus* subsp. *massiliense* | Rough | >16 | Resistant | >16 | Resistant | >16 | Resistant | Acquired | >128 | 128 | 64 | >128 | 128 |
| 80824 | *M. abscessus* subsp. *massiliense* | Mixed | 0.25 | Susceptible | 0.5 | Susceptible | 1 | Susceptible | Susceptible | 128 | 16 | 32 | 128 | 128 |
| 80838 | *M. abscessus* subsp. *abscessus* | Mixed | 1 | Susceptible | 16 | Resistant | 16 | Resistant | Inducible | 128 | 128 | 128 | 128 | >128 |
| 80866 | *M. abscessus* subsp. *massiliense* | Smooth | 0.12 | Susceptible | 0.12 | Susceptible | >16 | Resistant | Inducible | 128 | 128 | 128 | 128 | 128 |
| 80901 | *M. abscessus* subsp. *abscessus* | Rough | 1 | Susceptible | 8 | Resistant | 16 | Resistant | Inducible | >128 | 128 | 128 | 128 | >128 |
| 80988 | *M. abscessus* subsp. *abscessus* | Rough | 0.12 | Susceptible | 0.25 | Susceptible | 16 | Resistant | Inducible | 128 | >128 | 64 | 128 | >128 |
| 81103 | *M. abscessus* subsp. *abscessus* | Smooth | 2 | Susceptible | 2 | Susceptible | 16 | Resistant | Inducible | 128 | 128 | 64 | 128 | 128 |
| 81350 | *M. abscessus* subsp. *massiliense* | Rough | ≤0.06 | Susceptible | 0.12 | Susceptible | 0.12 | Susceptible | Susceptible | >128 | 128 | 128 | 128 | 128 |
| 81422 | *M. abscessus* subsp. *abscessus* | Rough | 0.5 | Susceptible | 1 | Susceptible | ≥16 | Resistant | Inducible | >128 | 128 | 128 | 128 | >128 |
| 81463 | *M. abscessus* subsp. *massiliense* | Smooth | ≤0.06 | Susceptible | 0.12 | Susceptible | 0.25 | Susceptible | Susceptible | 128 | 128 | 64 | 128 | 128 |
| 81499 | *M. abscessus* subsp. *massiliense* | Rough | 16 | Resistant | 16 | Resistant | 16 | Resistant | Acquired | 128 | 128 | 64 | 128 | 128 |
| 81618 | *M. abscessus* subsp. *abscessus* | Mixed | 16 | Resistant | 16 | Resistant | 16 | Resistant | Acquired | 128 | 16 | 32 | 128 | 128 |

*(Continued)*

| Isolates | Organism | Colony morphology | MIC value of clarithromycin (µg/mL) | | | | | | | Antimycobacterial screening against *M. abscessus* isolates | | | | |
|---|---|---|---|---|---|---|---|---|---|---|---|---|---|---|
| | | | Day 3 | | Day 5 | | Day 14 | | DST interpretation | MIC value of pure compounds (µg/mL) | | | | |
| | | | MIC | Phenotype | MIC | Phenotype | MIC | Phenotype | | RL008 | RL009 | RL011 | RL012 | RL013 |
| 81652 | *M. abscessus* subsp. *massiliense* | Mixed | 0.12 | Susceptible | 0.25 | Susceptible | 1 | Susceptible | Susceptible | 128 | 8 | 128 | 64 | 128 |
| 81702 | *M. abscessus* subsp. *massiliense* | Rough | >16 | Resistant | >16 | Resistant | >16 | Resistant | Acquired | >128 | 128 | 64 | >128 | 128 |
| 81838 | *M. abscessus* subsp. *massiliense* | Rough | 16 | Resistant | 16 | Resistant | ≥16 | Resistant | Acquired | 128 | 128 | 128 | 128 | 128 |
| 82119 | *M. abscessus* subsp. *abscessus* | Smooth | 2 | Susceptible | 16 | Resistant | 16 | Resistant | Inducible | 128 | 16 | 64 | 64 | 128 |
| 82154 | *M. abscessus* subsp. *abscessus* | Smooth | 0.5 | Susceptible | 8 | Resistant | 16 | Resistant | Inducible | 128 | 32 | 64 | 64 | 128 |
| 82593 | *M. abscessus* subsp. *massiliense* | Rough | ≤0.06 | Susceptible | 0.12 | Susceptible | 0.5 | Susceptible | Susceptible | >128 | 128 | 128 | >128 | >128 |
| 82895 | *M. abscessus* subsp. *massiliense* | Rough | 0.25 | Susceptible | 0.5 | Susceptible | 2 | Susceptible | Susceptible | 128 | >128 | 32 | 128 | >128 |
| 82998 | *M. abscessus* subsp. *abscessus* | Smooth | 16 | Resistant | 16 | Resistant | 16 | Resistant | Acquired | 128 | 128 | 128 | 128 | 128 |
| 83305 | *M. abscessus* subsp. *abscessus* | Smooth | 16 | Resistant | 16 | Resistant | 16 | Resistant | Acquired | >128 | 128 | 128 | 128 | 128 |
| 83310 | *M. abscessus* subsp. *abscessus* | Rough | 8 | Resistant | 8 | Resistant | ≥16 | Resistant | Acquired | 128 | >128 | 128 | >128 | >128 |
| 83347 | *M. abscessus* subsp. *massiliense* | Mixed | 0.25 | Susceptible | 0.5 | Susceptible | 0.5 | Susceptible | Susceptible | 128 | 128 | 128 | 128 | >128 |
| 90919 | *M. abscessus* subsp. *massiliense* | Mixed | 0.12 | Susceptible | 0.12 | Susceptible | 0.5 | Susceptible | Susceptible | 128 | 128 | 128 | 128 | >128 |
| 826492 | *M. abscessus* subsp. *massiliense* | Mixed | 0.12 | Susceptible | 0.25 | Susceptible | 16 | Resistant | Inducible | 128 | 16 | 64 | 64 | 128 |

Note:
MIC, minimum inhibitory concentration; DST, drug susceptibility testing; Acquired, acquired resistance; Inducible, inducible resistance.

0.50), which inhibited eight *M. abscessus* isolates (57%) (Fig. 1). The second strongest synergistic activity was observed for the RL012/CLA combination, followed by the RL013/CLA combination, which showed synergistic effects against seven (50%) and six isolates (42.9%), respectively. The RL011/CLA combination showed the lowest synergistic effect, inhibiting only five isolates (35.7%).

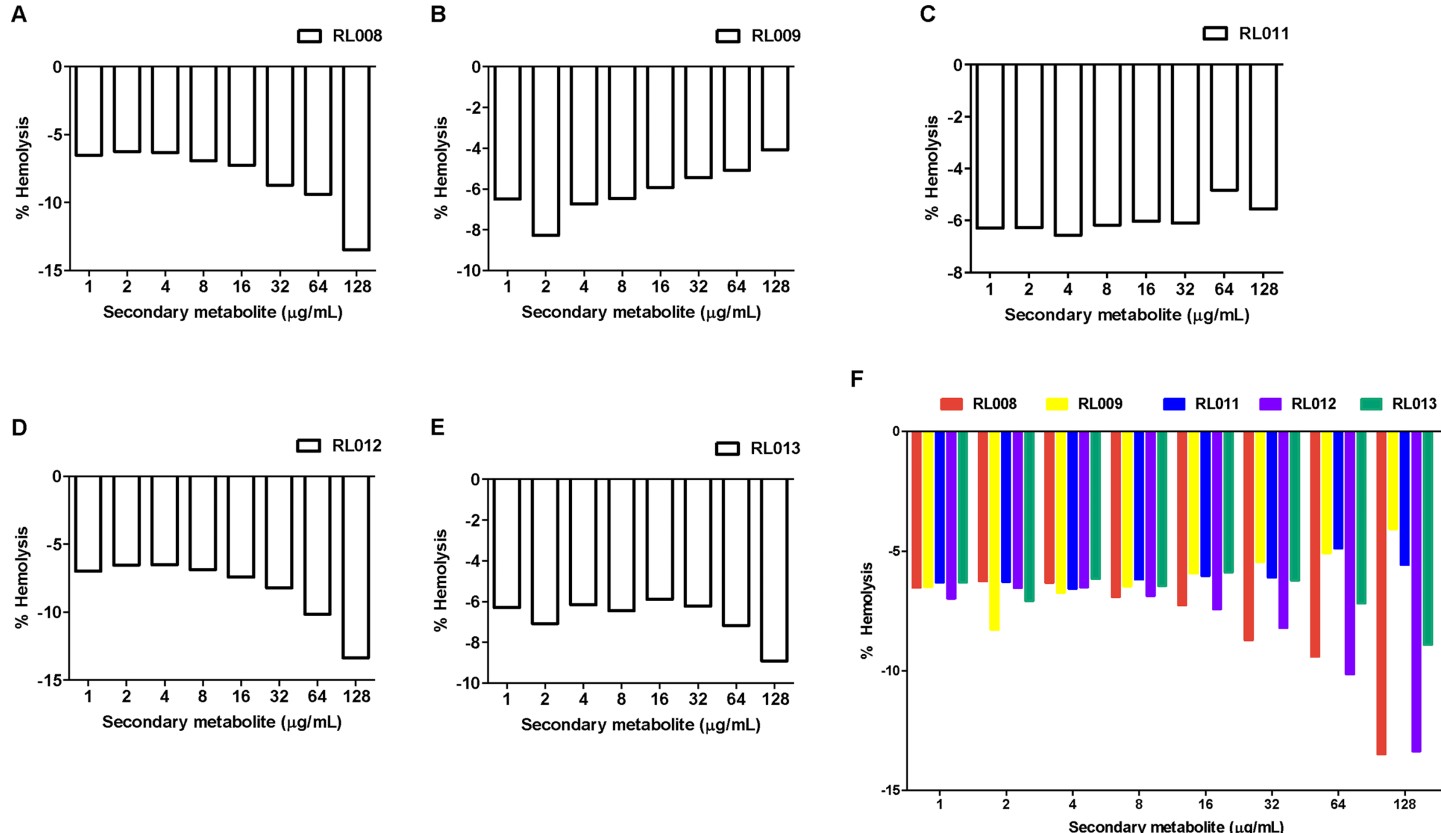

**Figure 2 Hemolytic activity of plant secondary metabolites against red blood cells (RBCs).** RBC suspension was incubated for 1 h with different concentrations of (A) RL008, (B) RL009, (C) RL011, (D) RL012 and (E) RL013. The hemolytic activity is presented as the percentage of hemolysis. Data are expressed as means. **Note:** When the % hemolysis of the tested concentration is lower than the control (negative value), 0% hemolysis is used as the baseline. Concentrations of secondary metabolites before adding RBC suspension are shown.

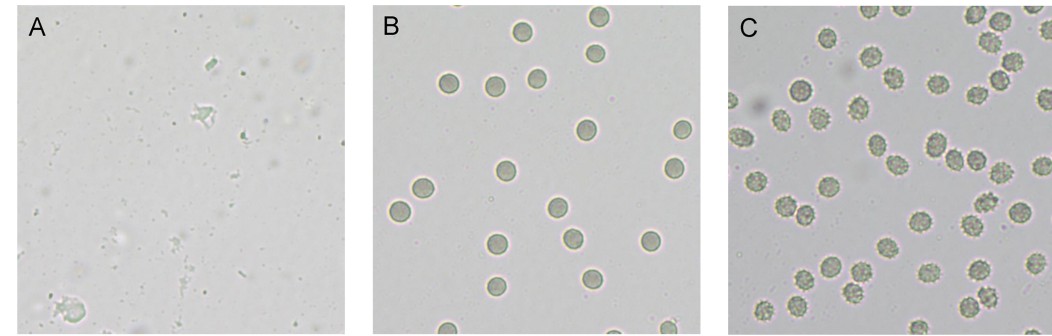

**Figure 3 Red blood cell (RBC) morphology under the light microscope.** RBC suspension was incubated for 1 h with (A) 1% Triton-X (positive control, showing complete hemolysis), (B) 1% PBS (negative control, showing no hemolysis) and (C) 128 μg/mL of RL013 (showing some shrinkage due to osmotic effects). With the remaining four compounds, RBCs exhibited similar morphology of cell shrinkage, especially at high MICs.

## DISCUSSION

*Mycobacterium abscessus* mostly occurs as a chronic infection and is an important cause of morbidity and mortality (*Cassidy et al., 2009*). The emergence, evolution and spread of

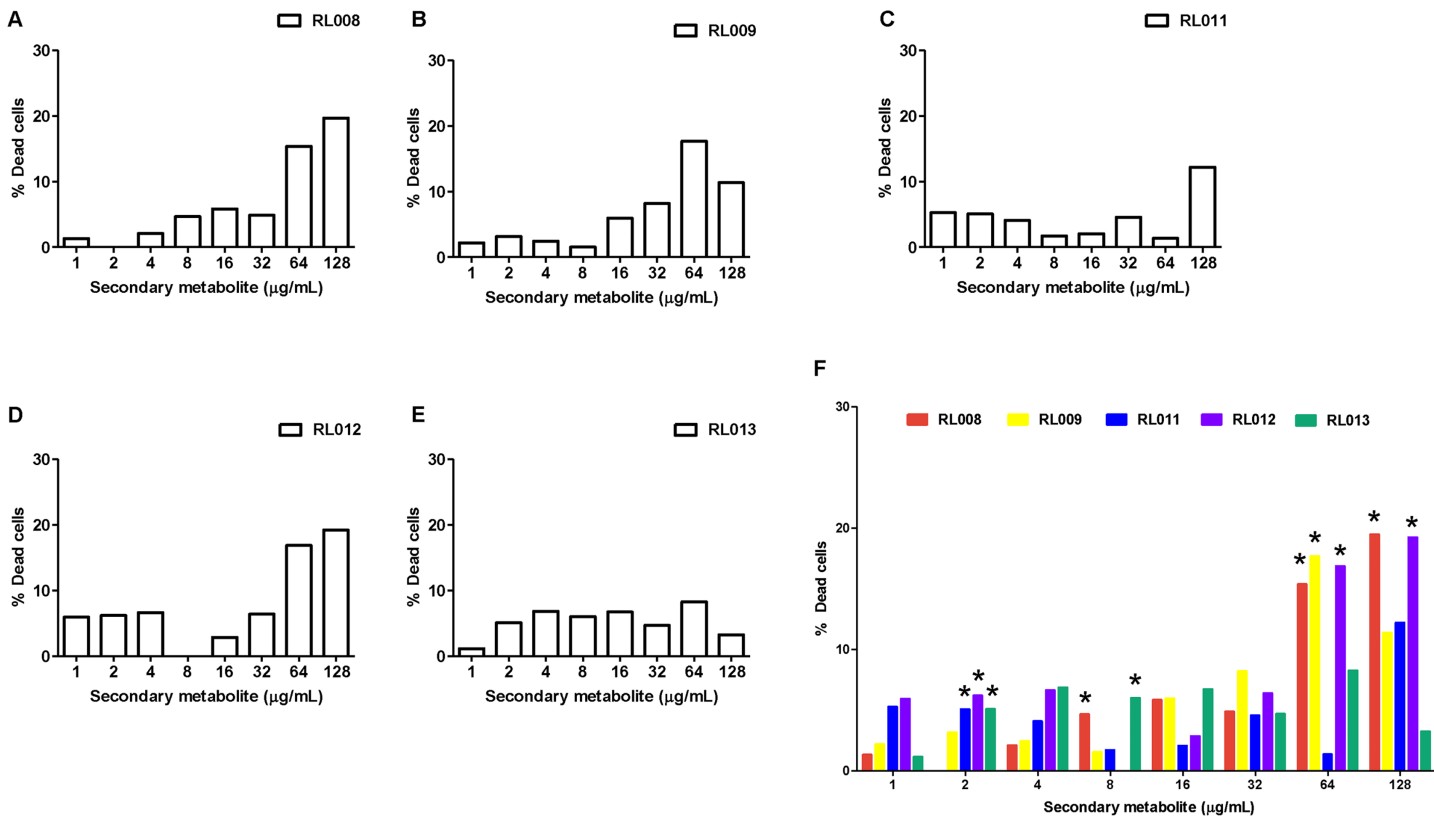

**Figure 4 Effect of plant secondary metabolites on white blood cells (WBCs).** WBCs were incubated for 1 h with different concentrations of (A) RL008, (B) RL009, (C) RL011, (D) RL012, (E) RL013 and (F) One-way ANOVA followed by post-hoc LSD test was used to determine significant differences ($^*P < 0.05$). The toxicity of each secondary metabolite is expressed as the percentage of dead cells. Data are expressed as means.

*M. abscessus* infection is highly associated with drug resistance and treatment failure (*Tung et al., 2015*). Antibiotic resistance to *M. abscessus* is a major public health concern worldwide, including in Thailand (*Imwidthaya et al., 1990*; *Phowthongkum et al., 2005*). Clarithromycin, a macrolide antibiotic, has a broad spectrum of antimicrobial activity that inhibits a range of Gram-positive and Gram-negative microorganisms (*Peters & Clissold, 1992*). It is often a drug of choice for the treatment of serious infections caused by *M. abscessus*. However, *M. abscessus* clinical isolates with reduced susceptibility to clarithromycin have emerged, resulting in a prolonged treatment course and poor clinical outcomes (*Li et al., 2017*). Clarithromycin monotherapy is associated with treatment failure. A combination of antimicrobial agents may be of therapeutic benefit and efficacious in the treatment of infections caused by clarithromycin-resistant *M. abscessus*. There is therefore a need to search for new sources of antimycobacterial substances. Plants produce a variety of bioactive compounds, sometimes with known therapeutic properties (*Rios & Recio, 2005*). They are good sources of powerful antibiotic metabolites and can treat various diseases (*Hernandez-Garcia et al., 2019*). This study was conducted to evaluate the antimycobacterial activity of different secondary metabolites of plant origin against clarithromycin-susceptible and-resistant *M. abscessus* isolates.

**Table 3 Synergistic activity of five plant secondary metabolites combined with clarithromycin against 14 clarithromycin-resistant *M. abscessus* isolates.**

| Isolates | Susceptibility profile | Individual MIC (μg/mL) | | | | | | Combination MIC (μg/mL) | | | | | FICI | | | | |
|---|---|---|---|---|---|---|---|---|---|---|---|---|---|---|---|---|---|
| | | CLA | RL008 | RL009 | RL011 | RL012 | RL013 | CLA/RL008 | CLA/RL009 | CLA/RL011 | CLA/RL012 | CLA/RL013 | RL008 | RL009 | RL011 | RL012 | RL013 |
| 81499 | Acquired | 64 | 128 | 128 | 64 | 128 | 128 | 4/16 | 32/128 | 64/128 | 4/8 | 16/64 | 0.19 | 1.50 | 2.00 | 0.13 | 0.75 |
| 82998 | Acquired | 16 | 128 | 128 | 128 | 128 | 128 | 8/32 | 2/32 | 8/64 | 8/128 | 8/32 | 0.75 | 0.38 | 1.00 | 1.50 | 0.75 |
| 83305 | Acquired | 16 | >128 | 128 | 128 | 128 | 128 | 16/128 | 2/128 | 16/128 | 4/128 | 4/64 | 2.00 | 1.13 | 2.00 | 1.25 | 0.75 |
| 83310 | Acquired | 4 | 128 | >128 | 128 | >128 | >128 | 1/8 | 2/8 | 1/8 | 4/8 | 1/8 | 0.31 | 0.56 | 0.31 | 1.06 | 0.31 |
| 81618 | Acquired | 64 | 128 | 16 | 32 | 128 | 128 | 64/128 | 32/64 | 64/128 | 64/128 | 64/128 | 2.00 | 1.00 | 2.00 | 2.00 | 2.00 |
| 80700 | Acquired | 512 | >128 | 128 | 64 | >128 | 128 | 512/128 | 512/128 | 512/128 | 512/128 | 512/128 | 2.00 | 2.00 | 2.00 | 2.00 | 2.00 |
| 81838 | Acquired | 16 | 128 | 128 | 128 | 128 | 128 | 1/8 | 1/8 | 1/8 | 1/8 | 1/8 | 0.13 | 0.13 | 0.13 | 0.13 | 0.13 |
| 82119 | Inducible | 512 | 128 | 16 | 64 | 64 | 128 | 32/8 | 512/128 | 512/64 | 64/128 | 512/64 | 0.50 | 2.00 | 1.50 | 1.13 | 1.5 |
| 80097 | Inducible | 512 | 128 | 16 | 64 | 64 | 128 | 32/8 | 64/32 | 32/64 | 64/32 | 128/64 | 0.13 | 0.38 | 0.56 | 0.38 | 0.75 |
| 80838 | Inducible | 128 | 128 | 128 | 128 | 128 | >128 | 32/8 | 32/16 | 64/8 | 16/16 | 16/32 | 0.31 | 0.38 | 0.56 | 0.25 | 0.38 |
| 80988 | Inducible | 4 | 128 | >128 | 64 | 128 | >128 | 2/8 | 1/32 | 1/8 | 2/8 | 2/8 | 0.56 | 0.50 | 0.31 | 0.56 | 0.56 |
| 80901 | Inducible | 16 | >128 | 128 | 128 | 128 | >128 | 4/8 | 4/8 | 4/8 | 4/8 | 4/16 | 0.31 | 0.31 | 0.31 | 0.31 | 0.38 |
| 82154 | Inducible | 32 | 128 | 32 | 64 | 64 | 128 | 4/16 | 8/16 | 8/16 | 8/32 | 8/16 | 0.25 | 0.38 | 0.38 | 0.50 | 0.38 |
| 81103 | Inducible | 64 | 128 | 128 | 64 | 128 | 128 | 32/8 | 16/32 | 16/128 | 8/32 | 16/32 | 0.56 | 0.50 | 1.25 | 0.38 | 0.50 |
| Numbers of isolates with synergistic activity (%) | | | | | | | | | | | | | 8 (57) | 8 (57) | 5 (35.7) | 7 (50) | 6 (42.9) |

**Notes:**

MIC, minimum inhibitory concentration; CLA, clarithromycin; Acquired, acquired resistance; Inducible, inducible resistance; FICI, fractional inhibitory concentration index.
FICI interpretation: ≤0.5: synergy; >0.5–1.0: additive; >1.0–4.0: indifference; >4.0: antagonism (*Doern, 2014*).
Grey-shaded boxes shows synergistic effect.

Researchers have isolated several such compounds and demonstrated their activities against mycobacteria, including *M. tuberculosis* (*Baldin et al., 2018*; *Chaipukdee et al., 2016*; *Jimenez-Arellanes et al., 2013*; *Jyoti et al., 2016*; *Kanokmedhakul et al., 2012*; *Molina-Salinas et al., 2006*; *Naik et al., 2014*; *Nkot et al., 2018*). However, antimycobacterial activities of secondary metabolites against *M. abscessus* have rarely been reported. We used 23 secondary metabolites isolated from *A. monophylla*, *P. filamentosa*, *A. conyzoides*, *R. wittii* and *N. nambi* against *M. abscessus* with different clarithromycin-resistance levels. The most effective compounds were RL008, RL009, RL011, RL012, and RL013, which exhibited MIC values ranging from <1 to >128 μg/mL.

Previous reports showed that colony morphology was not associated with susceptibility to first-line antibiotics (*Ruger et al., 2014*). However, *Clary et al. (2018)* reported that certain colony morphotypes of *M. abscessus* were associated with biofilm formation and prolonged intracellular survival. We investigated the relationship between the colony morphotypes and the susceptibility to the compounds and clarithromycin. No significant association was found. This might be due to inadequate

sample size. Therefore, such a relationship is still unconfirmed and requires further investigation.

Checking the toxicity of secondary metabolites on both human RBCs and WBCs is of importance when selecting candidates for antimycobacterial drugs. Our results demonstrate that the selected secondary metabolites are not harmful towards RBCs. No hemolysis was found among various concentrations of each compound. In fact, the degree of hemolysis in the presence of the test compounds was lower than in the negative controls (PBS), indicating the compounds may help to preserve RBCs better than the PBS control. The RBC shrinkage observed at high compound concentrations was an osmotic effect due to the extra-cellular concentrations. These results agree with those from a previous study (*Lima Viana et al., 2018*), which evaluated the antimicrobial activity of *Bixa orellana* secondary metabolites to treat *Mycobacterium* infections. In our study, the secondary metabolites tested did not induce significant toxicity in human RBCs. The plant secondary metabolites that we tested killed 3% to 20% of WBCs at their MIC levels (128 µg/mL). RL013 had the lowest cytotoxicity to leukocytes with the fewest WBC deaths (≈3%), lower than that caused by RL009 and RL012. Although toxicity for WBCs was quite high, these compounds nevertheless had potential to inhibit clarithromycin-resistant *M. abscessus* isolates. While chemical hair dye had similar toxicity for human WBCs and RBCs (*Maiti et al., 2016*), our plant-extracted compound had higher toxicity for WBCs than for RBCs. The ability of immune cells (*i.e.*, WBCs) to absorb foreign compounds might be a possible explanation (*Keselowsky, Acharya & Lewis, 2020*; *Wang et al., 2018*). Therefore, compounds with less toxicity for leukocytes should be selected for further study to avoid affecting the host immune system.

In this study, a combination of the tested compounds and clarithromycin had synergistic effects on some *M. abscessus* isolates with acquired or inducible clarithromycin resistance. No antagonistic effect of combining these substances was found. Among the five effective compounds tested, RL008 and RL009 proved to be the best in a combined treatment with clarithromycin, frequently showing a synergistic effect, with the FICI values ranging from 0.13 to 0.5. Combined with RL008 and RL009, the average MICs of clarithromycin alone were reduced up to 16-fold (*i.e.*, reduced from 512 to 32 µg/mL). Both RL008 and RL009 had low toxicity against RBCs and WBCs at the MIC levels. These results are consistent with those of *Rahgozar, Bakhshi Khaniki & Sardari (2018)*, who found that the best synergistic results against *Mycobacterium bovis* were obtained for extracts of *Lavandula stoechas* and *Datura stramonium* in combination with ethambutol. In addition, *Lopes et al. (2014)* reported that a synergism was observed against *M. tuberculosis* with eupomatenoid-5 (EUP-5), extracted from *Piper solmsianum* C. DC. var. *solmsianum* plus rifampicin, and EUP-5 plus ethambutol combinations. Similar observations of synergism between plant secondary metabolites and various drugs against mycobacteria have been reported in other studies (*Aro et al., 2016*; *Mossa, El-Feraly & Muhammad, 2004*; *Naik et al., 2014*). We observed that synergistic effects of combining the test compounds with clarithromycin occurred against *M. abscessus* isolates with inducible as well as acquired

clarithromycin resistance. Therefore, plant secondary metabolites could be used for treatment of both forms of resistance. However, the association between antimycobacterial susceptibility and clarithromycin-resistance type remains unclear.

Nowadays, the frequent treatment failure of *M. abscessus* infection is a major public health concern. Although our combinations of pure compounds and clarithromycin did not exhibit synergistic effects against all isolates, almost 60% of clarithromycin-resistant *M. abscessus* isolates (showing either inducible or acquired resistance) were inhibited. This information might be applied in development of alternative treatments for *M. abscessus* infection.

The core structure of both RL008 and RL009 is an anthraquinone, an aromatic organic compound. This study supports previous findings that anthraquinone compounds possess antimicrobial properties against Gram-positive bacteria, Gram-negative bacteria, and fungi (*Comini et al., 2011*; *Kemegne et al., 2017*; *Lu et al., 2011*; *Xu et al., 2017*).

Limitations of this study should be acknowledged. Toxicity tests were performed only for RBCs and WBCs. However, these secondary metabolites that we tested *in vitro* should be further investigated *in vivo* for more conclusive results. The relationship between the colony morphotype of *M. abcessus* isolates and their susceptibility to the combination of compounds and clarithromycin is required. Further work is necessary using structural variants of the plant secondary metabolites identified here to improve their antimycobacterial efficacy. Additional studies might also evaluate antifungal, antiviral and antiparasitic activities of these compounds.

## CONCLUSION

We report that five compounds isolated from medicinal plants have potent antimycobacterial effects, which are enhanced synergistically when combined with clarithromycin against clarithromycin-resistant *M. abscessus* clinical isolates. They also showed acceptable results in toxicity tests towards RBCs and WBCs. These compounds might be used as an alternative treatment and should be further studied to develop anti-tuberculous drugs.

## ACKNOWLEDGEMENTS

We would like to acknowledge Prof. David Blair for editing the MS *via* Publication Clinic KKU, Thailand.

### Funding

This study was financially supported by the Faculty of Medicine, Khon Kaen University, Thailand (Grant Number IN62308) and the Research and Diagnostic Center for Emerging Infectious Diseases (RCEID), Khon Kaen University, Thailand. The funders had no role in study design, data collection and analysis, decision to publish, or preparation of the manuscript.

## Grant Disclosures

The following grant information was disclosed by the authors:
Faculty of Medicine, Khon Kaen University, Thailand: IN62308.
Research and Diagnostic Center for Emerging Infectious Diseases (RCEID), Khon Kaen University, Thailand.

## Competing Interests

The authors declare that they have no competing interests.

## Author Contributions

- Auttawit Sirichoat performed the experiments, analyzed the data, prepared figures and/or tables, authored or reviewed drafts of the paper, and approved the final draft.
- Irin Kham-ngam performed the experiments, analyzed the data, prepared figures and/or tables, authored or reviewed drafts of the paper, and approved the final draft.
- Orawee Kaewprasert performed the experiments, analyzed the data, prepared figures and/or tables, and approved the final draft.
- Pimjai Ananta analyzed the data, authored or reviewed drafts of the paper, and approved the final draft.
- Awat Wisetsai analyzed the data, prepared figures and/or tables, and approved the final draft.
- Ratsami Lekphrom analyzed the data, authored or reviewed drafts of the paper, and approved the final draft.
- Kiatichai Faksri conceived and designed the experiments, performed the experiments, analyzed the data, prepared figures and/or tables, authored or reviewed drafts of the paper, and approved the final draft.

## Human Ethics

The following information was supplied relating to ethical approvals (*i.e.*, approving body and any reference numbers):
This study was approved by the Khon Kaen University Ethics Committee for Human Research (HE611496).

## DNA Deposition

The following information was supplied regarding the deposition of DNA sequences:
The raw sequences are available at NCBI GenBank: PRJNA523980.

## Data Availability

The raw data is available in the Supplementary Files.

## Supplemental Information

Supplemental information for this article can be found online at http://dx.doi.org/10.7717/peerj.12391#supplemental-information.

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
