# Peer review of "Assessment of antimycobacterial activities of pure compounds extracted from Thai medicinal plants against clarithromycin-resistant Mycobacterium abscessus"

_PeerJ, doi:10.7717/peerj.12391_

## Round 0.1 · original submission · Minor Revisions

Three experts in this field have reviewed your manuscript and found merits for publication in this journal. Please address the minor points raised by them.

Reviewer 1 ·

Basic reporting

In general, the research is carried out in a good way and the methods used are correct to achieve the stated objectives.

Experimental design

I consider that research has scientific contributions, especially in the treatment of microorganisms that are constantly acquiring resistance to conventional antibiotics.

Some considerations:

1.- Briefly explain why the toxicity tests of the compounds were performed on RBC and WBC and not on some other tissue.

2.- Is there any explanation or reference to why the compounds have a higher percentage of toxicity in WBC compared to RBC?

Validity of the findings

Line 269-272.
I suggest you discuss a little more about this topic and the results obtained that are shown in figure 1. And with that to reinforce whether or not there is a relationship between the morphology of the isolates and the susceptibility to the compounds and clarithromicin.

Line 284.
Why this reaction happens with the WBC and not with the RBC; is there any explanation or is there something in the literature about it?

This is important because of its role in the immune response.


Fig. 3. Place the measuring bar.



Line. 280. ...human RBCsThe plant --> human RBCs. The...


LIne 68.
Complete the medicinal effects of these plants. Is it known in the literature against which organisms they act?

Additional comments

Consider the suggestions displayed.

·

Basic reporting

The authors have investigated the antibacterial activity of several plant compounds and their effect cytotoxic effect. The compounds have a strong effect on various clarithromycin susceptible and resistant strains of M. abscessus. The study is well designed. However, the introduction needs to be elaborated to put forth a clear and detailed literature review and its development in the field. The role of these plants in folk or cultural medicine and the previous research developments about them should be written with a conclusion of the significance of this study at the end of the introduction. There are technical mistakes in presenting the data. The figures need editing and error bars to be depicted with clarity. The literature references are lacking appropriateness with methodology and findings. Other than that the work is divided into appropriate sub-sections of coherent linkages. The discussion part needs to be more coherent with results, the discussion of the findings with respect to already findings in the literature would broaden its standing and scope.

Experimental design

The experimental design is OK, methods need to be presented in detail to clearly understand the steps performed in a particular method. For statistical analysis, biological or technical replicates are a must, which needs to be incorporated in the methodology section. There is a discrepancy between methods and results.

Validity of the findings

The findings have been validated with sufficient experimental methods, though technical or biological replicates need attention. Raw data needs to be shared in supplemental. The manuscript is written in good English with few typo errors. The discussion needs to be more coherent and elaborate to increase its significance.

Additional comments

There are certain main concerns, which need to be addressed. Other minor comments are listed in the attached file.

·

Basic reporting

Language and grammar:
The English language is fine, some paragraphs need to be rewritten in order to better understand the idea of the text. These are marked on the PDF.

Literature and background:
The information provided in the introduction is very complete and covers everything mentioned in the work. You need to organize some paragraphs and add some bibliographies. The references that are cited in the introduction are related to the subject of study, which gives greater validity to the work.
Regarding the discussion, they explain all the results they obtained at work. Each of the findings was analyzed and compared with other research articles. The literature included in this section is linked to the results, which gives more support to the work. It is only necessary to mention the Gram-negative and Gram-positive microorganisms that are inhibited by the action of clarithromycin and add some bibliographic citations that are missing to the discussion.

Structure, figures, and tables:
The structure of the article is good and follows the basic writing parameters for a research article.
Regarding the figures and tables, the titles and descriptions are accurate and are complete. However, it is necessary to organize Table 2, add the percentages that are analyzed in the results of the work. In the Figure 2 and 3 is necessary to add the statistical analyzes. In Figure 2 and 4 the graphs should indicate if there are statistically significant data. These are marked on the PDF files.

Experimental design

Research question:
The research question is well defined and congruent with the discussion and conclusions.

Rigorous investigation:
The approaches and methodologies used are appropriate to answer the research question. However, it would be interesting to do the study in animals. It would be interesting to see how invertebrate organisms such as Drosophila melanogaster, Galleria mellonella, Tenebrio molitor, among others, react to this type of compound. I also consider that characterizing the types of compounds would be interesting. Learn more about its structure. Develop pilot studies to determine whether or not these compounds can affect human beings.

Methods:
It is necessary to detail more some methodologies, and add bibliographic citations. These are marked on the PDF. A statistical analysis section is needed.

Validity of the findings

Underlying data:
The controls necessary to carry out the experiments were included.

Conclusions:
The conclusions mentioned in this work are related to the original research question.

Additional comments

In general, the research question and methodologies are interesting and appropriate, as well as the results obtained. In addition, they aptly discuss what was found. They also mention what their results mean and the importance of these. The bibliographic support they used is very good, since the references are related to the work. It is an interesting job. However, other experiments must be carried out to test the results in patients or other organisms.

---

## Round 0.2 · accepted · Accept

The manuscript was significantly improved, based on the Reviewers' comments.